# Synthetic Amphipathic Helical Peptide L-37pA Ameliorates the Development of Acute Respiratory Distress Syndrome (ARDS) and ARDS-Induced Pulmonary Fibrosis in Mice

**DOI:** 10.3390/ijms25158384

**Published:** 2024-08-01

**Authors:** Aleksandr S. Chernov, Georgii B. Telegin, Alexey N. Minakov, Vitaly A. Kazakov, Maksim V. Rodionov, Viktor A. Palikov, Anna A. Kudriaeva, Alexey A. Belogurov

**Affiliations:** 1Shemyakin and Ovchinnikov Institute of Bioorganic Chemistry of the Russian Academy of Sciences, Moscow 117997, Russia; telegin@bibch.ru (G.B.T.); minakov@bibch.ru (A.N.M.); vitalij.tomsk@list.ru (V.A.K.); vpalikov@bibch.ru (V.A.P.); anna.kudriaeva@gmail.com (A.A.K.); alexey.belogurov.jr@gmail.com (A.A.B.J.); 2Medical Radiological Research Center (MRRC) Named after A.F. Tsyb-Branch of the National Medical Radiological Research Center of the Ministry of Health of the Russian Federation, Obninsk 249031, Russia; mvrodionov@inbox.ru; 3Department of Biological Chemistry, Russian University of Medicine of the Ministry of Health of the Russian Federation, Moscow 127473, Russia

**Keywords:** mice, model ARDS/DAD, pulmonary inflammation, cytokines, peptide L-37pA

## Abstract

In this study, we evaluated the ability of the synthetic amphipathic helical peptide (SAHP), L-37pA, which mediates pathogen recognition and innate immune responses, to treat acute respiratory distress syndrome (ARDS) accompanied by diffuse alveolar damage (DAD) and chronic pulmonary fibrosis (PF). For the modeling of ARDS/DAD, male ICR mice were used. Intrabronchial instillation (IB) of 200 µL of inflammatory agents was performed by an intravenous catheter 20 G into the left lung lobe only, leaving the right lobe unaffected. Intravenous injections (IVs) of L-37pA, dexamethasone (DEX) and physiological saline (saline) were used as therapies for ARDS/DAD. L37pA inhibited the circulating levels of inflammatory cytokines, such as IL-8, TNFα, IL1α, IL4, IL5, IL6, IL9 and IL10, by 75–95%. In all cases, the computed tomography (CT) data indicate that L-37pA reduced lung density faster to −335 ± 23 Hounsfield units (HU) on day 7 than with DEX and saline, to −105 ± 29 HU and −23 ± 11 HU, respectively. The results of functional tests showed that L-37pA treatment 6 h after ARDS/DAD initiation resulted in a more rapid improvement in the physiological respiratory lung by 30–45% functions compared with the comparison drugs. Our data suggest that synthetic amphipathic helical peptide L-37pA blocked a cytokine storm, inhibited acute and chronic pulmonary inflammation, prevented fibrosis development and improved physiological respiratory lung function in the ARDS/DAD mouse model. We concluded that a therapeutic strategy using SAHPs targeting SR-B receptors is a potential novel effective treatment for inflammation-induced ARDS, DAD and lung fibrosis of various etiologies.

## 1. Introduction

Class B scavenger receptors (SR-Bs), including SR-BI, its splicing variant SR-BII, and CD36, are multifunctional receptors, mostly known for their critical role in lipoprotein metabolism [1,2,3]. These receptors have also been implicated in the innate host responses to various pathogen-associated molecular pattern molecules (PAMPs) [4,5,6,7,8] and damage-associated molecular pattern molecules (DAMPs) [9,10,11]. Recently, the role of SR-Bs in sepsis has been increasingly investigated, and several studies using SR-BI/-BII- and CD36-knockout mice demonstrated their protective role in cecal ligation and puncture (CLP)-induced sepsis and endotoxemia [12,13,14,15,16]. SAHPs demonstrated anti-inflammatory effects in models of atherosclerosis [17,18], cardiac ischemia–reperfusion injury, endothelial dysfunction [19,20,21], endotoxemia, sepsis [16] and collagen-induced arthritis [22].

An early study on acute lung injury (ALI) as well as studies utilizing intraperitoneal (IP) LPS injections in gain-of-function models of transgenic mice with human SR-BI (hSR-BI) and hSR-BII expression revealed that both receptors mediate LPS uptake and contribute to LPS-induced inflammatory signaling and tissue injury, including the liver and kidney [12], thus supporting a potentially detrimental role of these receptors during IP-induced endotoxemia and LPS interaction with the liver and kidney reticular endothelial system (RES) [23]. When the same transgenic animals were intratracheally instilled (IT) with LPS, alveolar epithelial cells demonstrated increased LPS uptake and neutralization during the acute phase of ALI at 6–24 h [4]. Bacterium- or virus-induced ALI, acute respiratory distress syndrome (ARDS) and diffuse alveolar damage (DAD) followed by PF are significant sources of morbidity and mortality, with no current treatments beyond supportive care [24,25,26]. It was previously reported that ALI can be treated by antagonists of SR-Bs, synthetic amphipathic helical peptides (SAHPs), such as IV-injected L-37pA and ELK-B, which resulted in reduced acute inflammation, phagocyte lung infiltration and pulmonary endothelial barrier dysfunction induced by IT LPS [27]. With the Toll-like receptor (TLR)-mediated signaling pathway being established as the major contributor to host responses in various acute inflammation-associated syndromes, e.g., bacterial-LPS-induced systemic shock [28], sepsis [29,30] and ALI [31,32,33,34], SR-B receptors are known to play a supportive role in facilitating PAMP/DAMP uptake and innate immune response in vitro and in vivo. The role of class B scavenger receptors in ALI induced by a pathogen injection has been reported [4,27,35]. Moreover, previous data demonstrating that CD36, SR-BI and SR-BII play important roles in LPS-induced ALI, as well as the ability of SAHPs [19,27,36,37] to target these receptors during ARDS, DAD and PF, have not been well established.

In this study, we firstly tested the synthetic amphipathic helical peptide L-37pA on the novel, recently reported [38] model of ARDS/DAD, eliminating lethality and indefinitely extending the observational period.

## 2. Results

### 2.1. Contrast CT Bronchoscopy

Initially, the unilateral solution delivery to the left lung was visualized by utilizing a contrast solution of Omnipaque 240 with a CT bronchogram. The efficacy of Omnipaque 240 delivery can be observed as a light white signal contrast in the left lung. A three-dimensional left lung structure was visible with CT imaging (Figure 1A). No contrast delivery to the right lung was consistently observed. The H&E sections demonstrated normal lung structure of the mouse right lung at 1 h after bronchoscopy (Figure 1B, two magnifications).

### 2.2. Blood Plasma Chemokine and Cytokine Levels

Bio-Plex Pro Mouse Cytokine Panel 33-Plex was used to measure the levels of chemokines and cytokines to assess ARDS/DAD-induced systemic inflammatory responses (Figure 2). Blood plasma was collected after 3 h post-ARDS/DAD induction. 

Our data suggest that during the first 3 h, almost all cytokines and chemokines were significantly elevated 4- to 50-fold in the plasma. The administration of L37pA demonstrated significant (2- to 10-fold) reductions in the following cytokines: TNFα, IFNγ, IL-1α, IL-4, IL-5, IL-6, IL-9, IL-10 and IL-12 (Figure 2), as well as chemokines including MIP-1α, RANTES, MCP-1, KC and G-CSF (Figure 2). Both L37pA and DEX demonstrated high efficiency in reducing systemic inflammation during unilateral ARDS/DAD. A potential mechanism of L-37pA’s effect is the reduction of early cytokine responses mediated by IL-6, IFNγ and TNFα, produced by LPS stimulatory effect. Decreased IL-6 levels were associated with subsequently diminished IL-6-dependent chemokines MIP-1α and MCP-1, which reduce monocyte and dendrocyte pulmonary infiltration, as well as monocyte–granulocyte transition. The downregulation of G-CSF levels might also decrease granulocyte recruitment.

Assessment of lung density and lung volume in treated versus untreated animals with ARDS/DAD.

Three-dimensional CT images of the volume of lungs and their average density was performed using a CT scanner 7, 14, 30 and 45 days after ARDS/DAD. The assessment of average LD using Hounsfield units and LV in MM^3^ provides important insights into the lungs’ functional state and reveals a time-course of lung recovery after ARDS (Figure 3).

The development of ARDS/DAD in all experimental groups showed symptoms reflecting the extent of tissue injury determined by CT. On day 7 after ARDS/DAD induction, all animals demonstrated significant increases in the left LD up to approximately 50 ± 23 HU, with slight decreases in the left LV (Figure 3A). DEX-treated animals demonstrated the greatest decrease in the left LV, from 248 ± 29 to 156 ± 21 mm^3^ (Figure 3A), and this dynamic was observed until day 14.

Only L37pA treatment at 6 h provided an insignificant increase in the left LD before −103 ± 19 HU when compared to saline, DEX or other L37pA-treated groups. However, on day 14, in all mice treated with L37pA, the left LD was significantly lower than in those treated with saline and DEX (Figure 3A). This dynamic continued until day 30 (Figure 3A). The right LD is the control value demonstrating minimal changes over time (Figure 3B). However, on day 14, the right LV was compensatory and increased up to 15–30% in all groups, without changes in LD (Figure 3B). Minimal changes in the right LV were observed in animals treated with L37pA for 6h and 24h after ARDS/DAD induction. At the same time, the left LV continued to drop below 200 mm^3^, with LD recovering to -100 HU in DEX-treated mice and to −20 HU in saline-treated mice. L37pA-treated mice demonstrated recovery of both left LD and LV to pre-ARDS/DAD levels.

On day 30, nearly complete recovery of the left LV and LD was observed in all mice treated with L-37pA, while the saline- and DEX-treated groups had increased LV levels and reduced left LD levels to −127/−151 HU, respectively. On day 45, the left LD and LV were nearly normalized in all ARDS/DAD groups (Figure 3A, left). However, in the DEX and saline groups the volume of the right lungs remained above normal values (918 ± 22 mm^3^ and 867 ± 19 mm^3^, respectively).

### 2.3. Measurements of Respiratory Parameters in ARDS/DAD Treated With L37pA

We observed the development of ARDS/DAD-impaired respiratory functions with reduced respiratory minute volume (RMV) and MEF, while the respiratory rate (RR) was increased at 7 and 14 days (Figure 4). These parameters demonstrated statistically significant improvements upon DEX and L-37pA treatments compared to saline, with recovery appearing earlier in L-37pA-treated mice (14 days). The L37pA treatment without ARDS/DAD had stable respiratory function from day 7 to day 45.

### 2.4. Histological Analyses of ARDS/DAD Lungs

ARDS/DAD was also characterized by perivascular (PV), peribronchial (PB), alveolar wall (AW) and alveolar lumen (AL) mononuclear cell infiltration (Figure 5A, Table 1). In intact animals, all studied indicators were assessed as 0 points.

As seen in Figure 5A, on day 7 after ARDS/DAD lungs demonstrated extensive left lung damage with areas of atelectasis (decline areas, DAs) with a score of 3.67, up to 80% reduced alveoli volume and the obliteration of most bronchioles (Appendix A).

A single dose of DEX reduced the atelectasis area (decline area) to a score of 1.75, with mildly reduced PB, PV, AW and AL mononuclear infiltration (Figure 5A, Table 1). Treatment with L-37pA at 0, 6 and 24 h demonstrated a greater reduction in mononuclear infiltration and DAs when compared with dexamethasone (Figure 5A, Table 1). PB monocyte infiltration after the injection of saline, DEX, L-37pA 0h, L-37pA 6 h or L-37pA at 24 h had scores of 3.17, 3.0, 2.5 and 2.25, respectively, while PV monocyte infiltration demonstrated scores of 3.84, 3.5, 3.25 and 3.0, respectively. L37pA also reduced AW and AL mononuclear accumulation, but not as profoundly as PB and PV when compared to ARDS groups receiving saline or DEX.

The ARDS/DAD saline-treated group demonstrated heavier signs of inflammation, including PB infiltration (2.5), PV infiltration (3.5), AW and AL monocyte infiltration (3.0), and areas of lung collapse (2.0), on day 45 (Figure 5 and Appendix A). After a single injection of DEX, we observed signs of residual inflammation with PV infiltration (2.0) and areas of lung collapse (1.0) (Table 1). All L37pA-treated groups did not display DAs by day 45 after ARDS/DAD. The left lung demonstrated normal ventilation with all segments involved in blood oxygenation. We did not observe peribronchial infiltration; the perivascular infiltration score was 1.25, alveolar wall infiltration was 1.0 and the alveolar lumen score was 0.75. (Figure 5A, Table 1).

### 2.5. Pulmonary Fibrosis

On day 45 after ARDS/DAD induction, the left PF was observed histologically after Masson staining (Appendix A). L37pA-treated ARDS/DAD at all tested doses demonstrated greatly reduced fibrosis (Figure 5B, left chart). The Kernogan’s index was reduced to a nearly normal level at 0 h in L37pA-treated mice, while it was relatively unchanged with DEX treatment. A statistically non-significant trend toward reduction was also observed with the 6 and 24 h injections of L37pA after ARDS/DAD (Figure 5B, right chart).

## 3. Discussion

SAHPs known as apoA-I mimetic peptides [17,39], which retain the beneficial effects of apoA-I, have a helical amphipathic structure enabling their interactions with class B scavenger receptors including CD36, SR-BI, and SR-BII5 [39,40]; formyl-peptide receptors [41]; and, potentially, class A scavenger receptors. SAHPs demonstrated anti-inflammatory effects in models of atherosclerosis [17,18], cardiac ischemia–reperfusion injury, endothelial dysfunction [19,20,21], endotoxemia, sepsis [16] and collagen-induced arthritis [22]. Another SAHP5A decreased inflammatory and immune responses in a house dust mite-mediated murine model of asthma [42]. Additionally, the SAHPs L37pA and ELK-B targeting SR-B receptors are effective inhibitors of inflammation induced by IT instillation of LPS leading to ALI [27,43]. However, SAHPs effects on the long-lasting consequences of ALI, such as ARDS/DAD, pulmonary function, and the development of PF, are not currently researched.

In this study, we used a novel mouse model of a severe unilateral lung ALI leading to ARDS/DAD [38] followed by PF at 45 days of ARDS/DAD (Figure 1B and Figure 5). Unlike existing models [44,45,46], our model is characterized by 100% animal survival and abrupt increases in plasma levels of cytokines and chemokines. Tracheal instillation of LPS and GC into both lobes is not compatible with mice survival, which makes our model critically important for long-term analysis of ARDS, DAD and PF [44,45,46]. We were the first to successfully use the contrasting method with three-dimensional CT imaging to evaluate the efficiency of IT instillations of inflammatory agents in ARDS/DAD models in mice (Figure 1A).

Here, we show that in our model of ARDS/DAD L37pA but not DEX effectively reduced plasma cytokine and chemokine levels, indicating that L37pA has better efficacy than DEX (Figure 2). Some studies report that the administration of dexamethasone to human ACE2-transgenic mice with acute respiratory distress syndrome after SARS-CoV-2 inoculation could effectively ameliorate disease progression [47,48]. However, unlike the infectious model, our “sterile” model has other mechanisms for the development of inflammation, which determines the most effective operation of L37pA.

There are several potential mechanisms underlying the protective effects of SAHPs in ARDS/DAD/ALI. First, a key step appears to be the ability of these peptides to bind to SR-Bs and reduce CD36-mediated pulmonary inflammation. But, as is known, an increase in the level of CD36 leads to the activation of macrophages and the development of fibrosis in the lungs in ARDS [49,50]. Importantly, our results showed prevention of the long-lasting consequences of ALI and DAD, such as chronic monocyte pulmonary infiltration (Figure 5A) and PF (Figure 5B). These changes are of particular importance since, according to recent reports, COVID-19-associated PF has been observed after viral recovery [51,52,53].

Second, L37pA can also potentially bind LPS, neutralizing LPS lipid A and preventing the proinflammatory and toxic effects of LPS [27]. However, neutralization effects are diminished in the presence of cations and they can be less significant than receptor specificity [40]. Furthermore, L37pA was effective at 0, 6 and 24 h after LPS/GC instillation. This high effectiveness of ARDS/DAD treatment at 24 h when LPS is significantly cleared in the lungs indicates that LPS neutralization is not needed for the anti-ARDS/DAD/HF effects of the peptide [4]. This observation is consistent with the early stage of ARDS/DAD, ALI and PF development reported before [27,33].

Third, peptides could affect the LPS-induced effects mediated through other receptors. Previous reports suggested that in addition to SR-B, SAHPs may bind to other receptors including formyl peptide receptors, LOX-1, class A scavenger receptors [54] and the TLR family. However, TNF-α and phorbol ester-induced inflammatory stimulation mediated by none of the SR-B preceptors was only marginally affected by SAHPs in various cells, including macrophages [55]. Furthermore, recent data suggested that SR-B is one of the most important receptors associated with pathogen uptake, bacterial endotoxic cellular cytotoxicity and acute and chronic kidney and lung injury [13,16,35]. Other available data also suggested that the repeated cycles of PAMP-induced inflammation and DAMP-induced tissue damage may be mediated through CD36 [56]. At the same time, independently of the specific mechanisms affecting lung inflammation, SAHPs were demonstrated as important tools that decrease pulmonary damage, ARDS, DAD and PF induced by PAMPs (LPS) and DAMPs [57,58].

An analysis of the left (experimental) and right (negative control) LD and LV over a 45 day period demonstrated that while the left lung density of saline- or DEX-treated samples was only normalized by 45 days, the left lung of L37pA-treated samples started to recover at 14–30 days (Figure 3A). Previously, we and others have not been able to observe longer-term consequences of ARDS/DAD with such fulminant lung pathology and DAs due to the 100% lethality when using such inflammatory PAMP IT instillation bilaterally [27,43]. Consistent with the observed improvement in LD and volume, L37pA treatment significantly improved pulmonary ventilation as early as day 14 after ARDS/DAD initiation (Figure 4). All respiratory parameters such as RR (A), TV (B), MEF (C) and MV (D) were better on day 14 when compared to saline or DEX treatments.

## 4. Conclusions

Using the original unilateral ARDS/DAD model on mice, we firstly demonstrated that intravenous injection of L37pA at various time points effectively reduced acute and chronic inflammation, alveolar infiltration, alveolar wall and vascular infiltration and dysfunction, and PF, as well as markedly improved pulmonary physiological functions and accelerated pulmonary functional recovery by 2–3 weeks. The results of this study suggest that SAHPs such as L-37pA can be utilized as effective potent treatments for ARDS/DAD syndrome preventing pulmonary fibrosis. These results identify L37pA as a potential compound that could be developed as an additional therapy to treat pulmonary inflammation and dysfunction, such as those with various pulmonary infections and inflammatory insults.

In summary, we demonstrated that the administration of L37pA attenuates and prevents the development of several damage-associated pulmonary syndromes in mice after IB instillation of LPS/GC. These results identify L37pA as a potential compound that could be developed as an additional therapy to treat pulmonary inflammation and dysfunction, such as those with various pulmonary infections and inflammatory insults.

## 5. Materials and Methods

### 5.1. Reagents

We used LPS from Salmonella enterica (Millipore Sigma, Saint Louis, MO, USA), α-galactosylceramide (GC) (Avanti Polar Lipids, Alabaster, AL, USA) and propofol (Hana Pharmaceutical, Co., Ltd., Seoul, Republic of Korea). We used Omnipaque 240 (iohexol) (GE Healthcare AS, Oslo, Norway). The L-37pA peptide was from Chinese Peptide Company Ltd. (Hangzhou, China) and was synthesized as reported previously [27,39]. Dexamethasone (DEX) was from CSPC Ouyi Pharmaceutical Co. (Shijiazhuang, China).

### 5.2. Compounds, Targeted Delivery and Lung Damage

ICR mice weighing 40.1 ± 1.85 g (~4 months old) were used in all experiments. Animals were housed under standard conditions in the accredited IBCh animal breeding facility (the Unique Research Unit Bio-Model of the IBCh, RAS; the Bioresource Collection—Collection of SPF-Laboratory Rodents for Fundamental, Biomedical and Pharmacological Studies, No. 075-15-2021-1067). ARDS/DAD was induced by a single instillation of 200 µL saline containing 0.5 mg/mL of both LPS and α-galactosylceramide (GC) into the left lung bronchus as reported previously [38,59]. The trachea was intubated with an intravenous 20G catheter directed into the left main bronchia. Propofol at a dose of 20 mg/kg was injected via IV for anesthesia shortly before the intubation of trachea. The animals were randomly assigned to five treatment groups, with twelve animals in each group (*n* = 12). For L-37pA in saline solution, 100 µL was injected by IV via the tail vein (10 mg/kg) at 0, 6 and 24 h after ARDS initiation. Dexamethasone (DEX, 5 mg/kg) and physiological saline (0.9% NaCl) were injected by IV at 0 h after ARDS initiation.

All experimental procedures with animals were approved by the Institutional Animal Care and Use Committee of the Shemyakin and Ovchinnikov Institute of Bioorganic Chemistry of the Russian Academy of Sciences, protocol No. 746/20, from 20 April 2020. This study was conducted in accordance with the Shemyakin and Ovchinnikov Institute Regulations on Animal Care and the experiments were performed in accordance with ARRIVE guidelines and the regulations of this committee. All methods were performed according to relevant guidelines and regulations. This study used no human specimens.

### 5.3. Experimental Groups

This study included several experimental groups designed to verify early and late therapeutic effects of a single dose of the SAHP L37pA, comparing its effects to dexamethasone. The time-course included simultaneous 6 and 24 h intravenous L37pA treatments (Table 2).

### 5.4. Conformation of Targeted Delivery of Compounds and Lung Damage Evaluation

A CT scanner MRS*CT/PET (MR Solution, Guildford, UK) was used to assess the instillation of LPS/GC into the lungs, produce images and evaluate the extent of lung damage. The scanning parameters were energy 40 kVp, exposure 100 msec, and current 1 mA with a step angle of 1°. During imaging, the animals were anesthetized with 3% isoflurane in air and maintained in a specialized holding container at 37 °C. The obtained CT images were processed utilizing VivoQuant v. 4.0 software (Invicro, London, UK). Unilateral instillation of LPS/GC was verified at the bronchial tree using contrast bronchography with Omnipaque contrast (GE Healthcare AS, Oslo, Norway).

The lung volume and the average density of the left (experimental) and right (control) lung tissues were measured after computing segmentation of these lobes using either the automatic mode (using −300 to −800 Hounsfield units as the cut-off density) or manual mode. The volume was expressed in mm^3^ with the density presented in Hounsfield units (HU ± SEM).

### 5.5. Chemokine and Cytokine Assays

Mouse EDTA plasma was collected at 3 h after ARDS initiation. A Bio-Plex Pro Mouse Cytokine Panel 33-Plex (BCA-1/CXCL13, CTACK/CCL27, ENA-78/CXCL5, Eotaxin/CCL11, Eotaxin-2/CCL24, Fractalkine/CX3CL1, GM-CSF, I-309/CCL1,IFN-γ, IL-1β, IL-2, IL-4, IL-6, IL-10, IL-16, IP-10/CXCL10, I-TAC/CXCL11,KC/CXCL1, MCP-1/CCL2, MCP-3/CCL7,MCP-5/CCL12, MDC/CCL22,MIP-1α/CCL3, MIP-1β/CCL4,MIP-2/CXCL2, MIP-3α/CCL20, RANTES/CCL5,MIP-3β/CCL19, SCYB16/CXCL16, SDF-1α/CXCL12,TARC/CCL17,TECK/CCL25, TNF-α) (BIO-RAD, Hercules, CA, USA) was used to measure plasma levels of chemokines and cytokines. Plasma samples diluted at 1:3 (50 μL) were incubated with magnetic beads, washed and then incubated with detecting antibodies followed by Strepto-avidin-Phycoerythrin (SA-PE) incubation, according to the kit manufacturer’s protocol. Data were obtained using a Luminex 200 analyzer and analyzed using xPONENT software v. 3.1 as recommended by the manufacturer’s manual.

### 5.6. Assessment of Lung Density and Lung Volume in ARDS/DAD Mice

Three-dimensional CT images of the left and right lung structure were collected utilizing MRS*CT/PET (MR Solution, GB). Average lung density (LD) utilizing HU and lung volume (LV, in mm^3^) were used to assess lung injury, DAD development and ARDS/DAD recovery.

### 5.7. Measurements of Respiratory Dysfunction in ARDS/DAD Treated with L37pA

Respiratory parameters were analyzed using a computerized Power-Lab 8/35 instrument, (PL3508), spirometry block (FE141) and respiratory adapter (MLT1L) at 7, 14, 30 and 45 days. Immobilized mice were allowed to relax for 2 min to become comfortable. An animal breathing mask was applied, and after the appearance of a clear signal the data were recorded on LabChart-7 for 10 s in triplicates. The data included respiration frequency (n/min); maximal expiratory flow, MEF (mL/s); tidal volume (mL), TV; minute volume (mL), MV; and calculated respiration rate × tidal volume (total ventilation, mL/min), TV.

### 5.8. Histology Examinations

Animals were euthanized on days 7 (50%) and 45 (50%) after the induction of ARDS/DAD. Lungs were filled with 10% solution of buffered formalin (pH = 7.3). Paraffin sections 4–5 μm thick were stained with Hematoxylin/Eosin (H&E) and were further examined using standard light microscopy with an AxioScope A1 microscope (Carl Zeiss, Munich, Germany). The microphotographs of histological preparations were obtained using Axiocam 305 color and ZEN 2.6 lite software (Carl Zeiss, Munich, Germany). Lung pathophysiology and morphology were evaluated for peribronchial and perivascular inflammatory cell infiltration and the presence of immune-competent cells in the intra-alveolar septa and in the lumen of pulmonary acini. Mononuclear infiltration and inflammation were assessed semi-quantitatively utilizing a 5-score scale [38,60]. The presence of collapsed lung tissue, transudate in the lumen of alveoli, and foci of necrosis, as well as perivascular (PVI), peribronchial (PBI), pulmonary fibrosis (PF) and decline areas (DAs), were also evaluated. Utilizing ZEN 2.6 lite (Carl Zeiss, Germany) software, the Kernohan’s index was measured in the left lung lobe as a ratio of the thickness of the vascular wall relative to the diameter of the corresponding vessel that characterizes pulmonary blood circulation.

### 5.9. Statistical Analysis

Data are presented as mean ± standard deviation. Differences between treatment and control groups were tested for significance using SigmaPlot software v. 12 (SYSTAT Software Inc., Berkshire, UK), using Student’s *t*-test. *p*-values less than 0.05 were considered statistically significant. The original contributions presented in this study are included in the article/Appendix A. Further inquiries can be directed to the corresponding author.

## Figures and Tables

**Figure 1 ijms-25-08384-f001:**
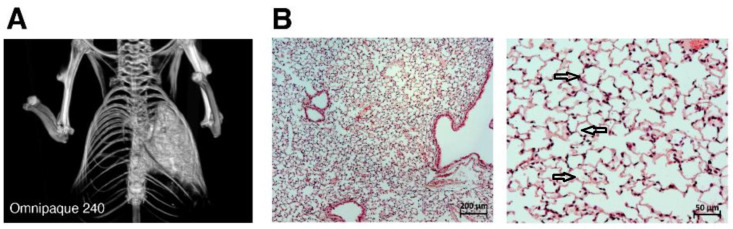
Lung analysis: (**A**) Three-dimensional CT image reconstruction (ventral view) of the left lung using the contrast agent Omnipaque 240. (**B**) Intact left lung histology analysis (magnification 50× and 200×); black arrows indicate normal alveoli.

**Figure 2 ijms-25-08384-f002:**
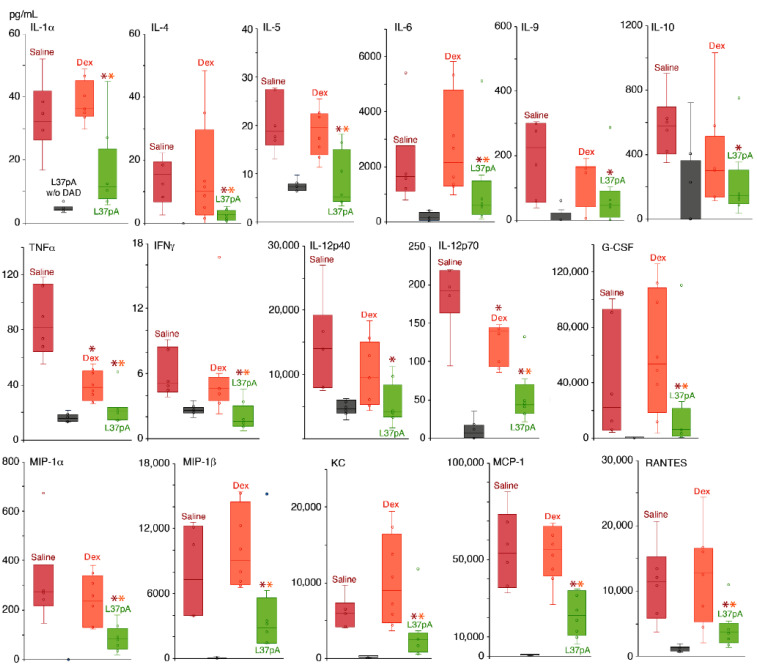
Levels of serum chemokines and cytokine in ICR mice with ARDS/DAD after administration of saline, DEX or L37pA. Asterisks denote statistically significant differences in saline (red) or dexamethasone (orange) test groups. Bars represent median, interquartile range with standard deviation (*p*-value: * < 0.05).

**Figure 3 ijms-25-08384-f003:**
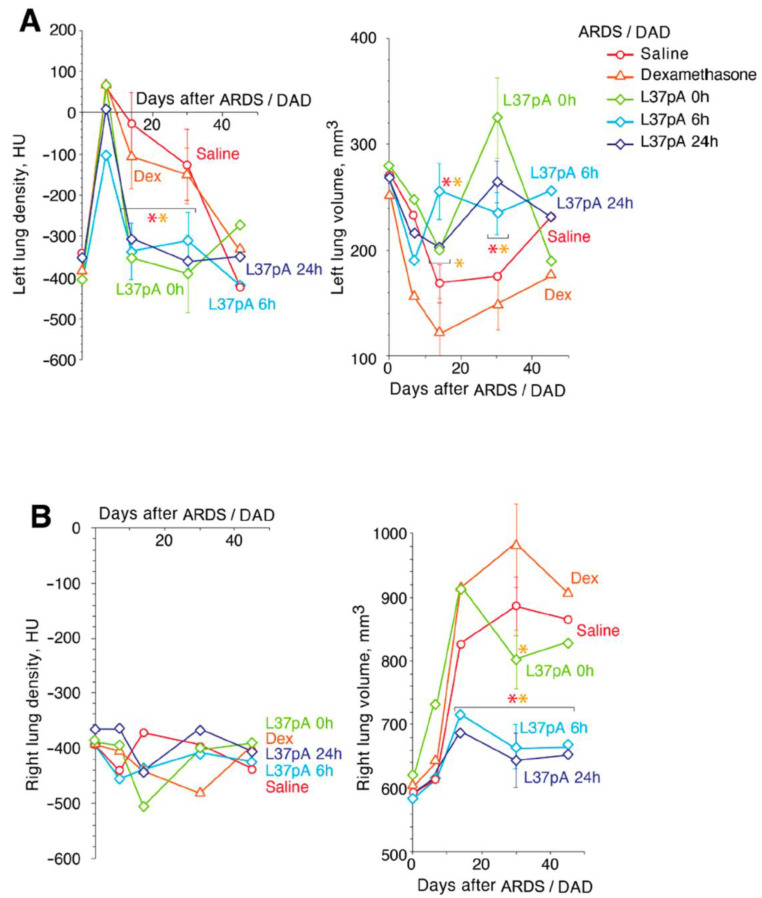
Treatment by L37pA induces fast recovery of the injured lung in ICR mice with ARDS/DAD. Density and volume of the **left** (**panel A**) and **right** (**panel B**) lung tissues in the experimental groups with ARDS/DAD. Asterisks denote statistically significant differences in saline (red) and dexamethasone (orange) test groups. Bars represent median, interquartile range with standard deviation (*p*-value: * < 0.05).

**Figure 4 ijms-25-08384-f004:**
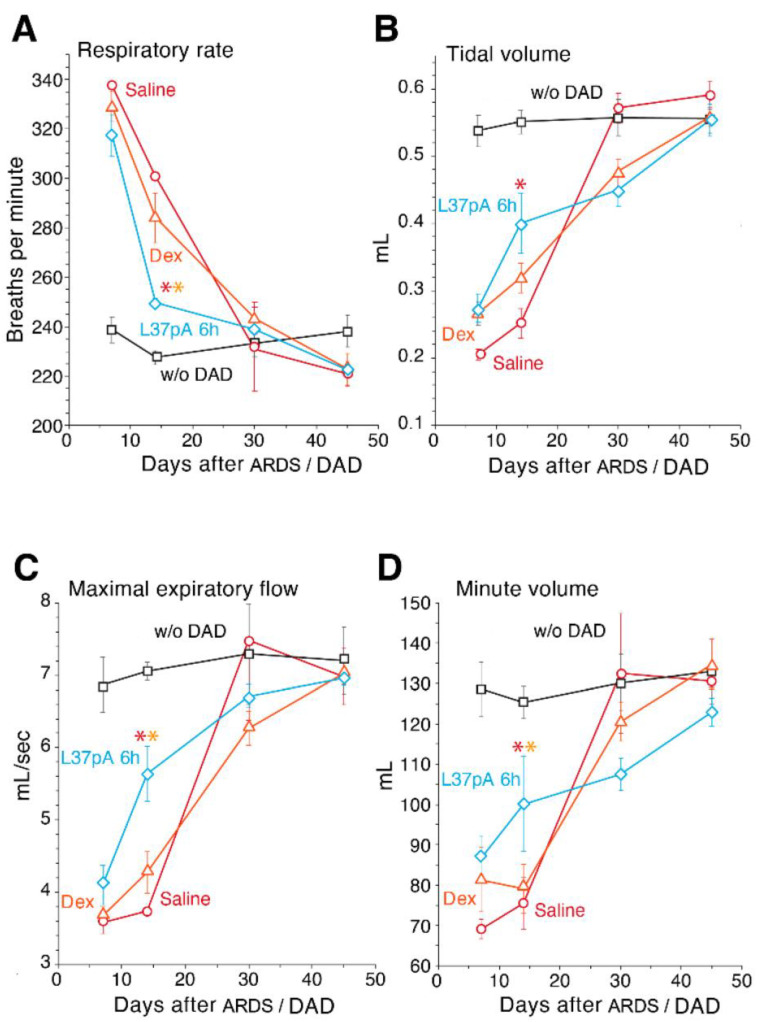
Measurement of respiratory function in mice with ARDS/DAD after administration by saline, DEX and L37pA. Respiratory rate (**panel A**); tidal volume (**panel B**); maximal expiratory flow (**panel C**) and minute volume (**panel D**) were assessed on day 7, 14, 30 and 45 after LPS/GC IB instillation and IV-injected saline, DEX, L37pA or in intact animals without DAD. Asterisks denote statistically significant differences in saline (red) or dexamethasone (orange) test groups. Bars represent median, interquartile range with standard deviation (*p*-value: * < 0.05).

**Figure 5 ijms-25-08384-f005:**
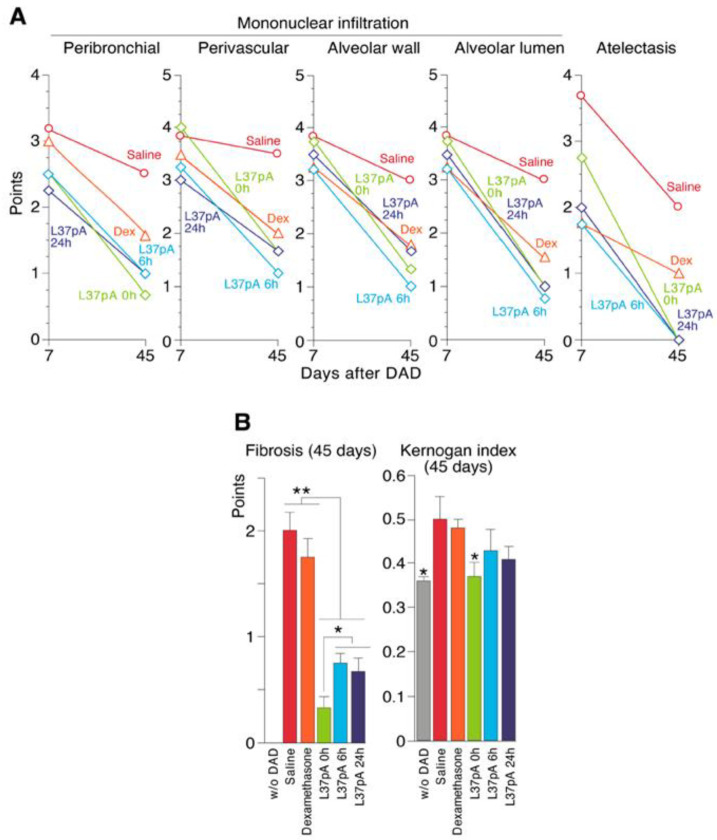
Left lung histology 7 and 45 days after ARDS/DAD treated with saline, DEX or L37pA. Results reflect a semi-quantitative analysis of lung injury in left lung. Hematoxylin/Eosin staining: left lung mononuclear infiltration (**panel A**); left lung fibrosis (Masson staining) on day 45 after ARDS/DAD initiation and Kernogan’s index (**panel B**). Statistically significant differences are indicated as *—*p* < 0.05; **—*p* < 0.005.

**Table 1 ijms-25-08384-t001:** Scoring scale of peribronchial (PB) and perivascular (PV) infiltration with a mononuclear cell, infiltration of alveolar walls (AWs) and ducts with the mononuclear cells and sites of pulmonary collapse (decline areas, DAs) of ICR mice 7 and 45 days after the induction of ARDS/DAD. 0—normal; 1—minimal expression; 2—mild; 3—medium; 4—strong; and 5—extraordinarily strong.

Days after ARDS/DAD	Groups	PB Infiltration	PV Infiltration	Mononuclear Infiltration AW	Mononuclear Infiltration AL	DA
7	ARDS + saline (*n* = 6)	3.17	3.84	3.84	3.84	3.67
	ARDS + IV DEX 0 h (*n* = 6)	3	3.5	3.25	3.25	1.75
	ARDS + IV 10 mg/mL L37pA at 6 h (*n* = 6)	2.5	3.25	3.25	3.25	1.75
	ARDS + IV 10 mg/mL L37pA at 24 h (*n* = 6)	2.25	3	3.5	3.5	2
	ARDS + IV 10 mg/mL L37pA at 0 h (*n* = 6)	2.5	4	3.75	3.75	2.75
45	ARDS + saline (*n* = 6)	2.5	3.5	3	3	2
	ARDS + IV DEX 0 h (*n* = 6)	1	2	1.75	1.5	1
	ARDS + IV 10 mg/mL L37pA at 6 h (*n* = 6)	1	1.25	1	0.75	0
	ARDS + IV 10 mg/mL L37pA at 24 h (*n* = 6)	1	1.67	1.67	1	0
	ARDS + IV 10 mg/mL L37pA at 0 h (*n* = 6)	0.67	1.67	1.33	1	0

**Table 2 ijms-25-08384-t002:** Experimental groups.

Group	ARDS by LPS + GC	Treatment	Group Description
1	ARDS	IV Saline	ARDS + Saline
2	ARDS	IV 5 mg/mL Dexamethasone	ARDS + DEX
3	ARDS	IV 10 mg/mL L37pA at 0 h	ARDS + L37pA at 0 h
4	ARDS	IV 10 mg/mL L37pA at 6 h	ARDS + L37pA at 6 h
5	ARDS	IV 10 mg/mL L37pA at 24 h	ARDS + L37pA at 24 h

## Data Availability

The original contributions presented in the study are included in the article/Appendix A. Further inquiries can be directed to the corresponding author.

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
