# Peer review of "Synthetic Amphipathic Helical Peptide L-37pA Ameliorates the Development of Acute Respiratory Distress Syndrome (ARDS) and ARDS-Induced Pulmonary Fibrosis in Mice"

_ijms, 2024, doi:10.3390/ijms25158384_

Round 1

Reviewer 1 Report

Comments and Suggestions for Authors

The paper is quite well written. The article covers a very interesting and current topic. Nevertheless, in my opinion, some parts need to be improved, I have some comments:

1) Abstract. Synthetic amphipathic helical peptides (SAHPs) can be used as the antagonists of class B scavenger receptors (SR-Bs), such as SR-BI, SR-BII, and CD36 which mediate pathogen recognition and innate immune responses. In this study, we evaluated the SAHP, L-37pA, to treat acute respiratory distress syndrome (ARDS) and the development of diffuse alveolar damage (DAD) and chronic pulmonary fibrosis (PF). For modeling of ARDS/DAD male ICR mice were used. Intrabronchially dose of inflammatory agents (LPS/GC) were performed by instilling into the left lung lobe only and leaving the right lobe unaffected. L-37pA was given intravenously (IV) simultaneously or 6 hours and 24 hours after the initiation of ARDS/DAD. Comparison drugs dexamethasone and physiological saline was IV injected at 0 hours after ARDS initiation. L37pA inhibited circulating levels of inflammatory cytokines including IL-8, TNFα, IL1α, IL4, IL5, IL6, IL9, and IL10 by 75%–95%. In all cases, on the CT data, L-37pA reduced lung density faster, what was manifested the reduced of lung edema. L-37pA treatment 6 hours after ARDS initiation resulted in a more rapid improvement of physiological respiratory lung functions when compared with comparison drugs.

Please, add some statistically significant data to support the results 

2) We conclude that L-37pA is a potent treatment for inhibiting acute and chronic pulmonary inflammation, dysfunction and lung fibrosis in ARDS/DAD. These data indicate that a therapeutic strategy using SAHPs targeting SR-B receptors is a potential novel effective treatment of inflammation-induced ARDS, DAD, and lung fibrosis of various etiologies.

Conclusion might be beneficial to include a sentence that briefly summarizes the key findings of the study. This can provide readers with a quick overview of the research. 

3) 1. Introduction. Please explain all the acronyms.

4) Utilizing this unilateral 68 ALI/ARDS/DAD model we have found that L-37pA, injected IV at various time points 69 effectively reduced acute and chronic inflammation, alveolar infiltration, alveolar wall 70 and vascular infiltration and dysfunction, PF, as well as markedly improved pulmonary 71 physiological functions and accelerated pulmonary functional recovery by 2–3 weeks. The 72 results of this study suggest that SAHPs such as L-37pA, can be utilized as effective potent 73 treatments of ARDS/DAD syndrome preventing PF. Please, improve the description of the study aim and underline the novelty of the paper.

5) 2. Results. Please underline the most important statistically significant data to support the results.

6) 3. Discussion 248 SAHP known as apoA-I mimetic peptides that retain the beneficial effects of apoA-I, 249 have been described earlier [37,38]. SAHP have a helical amphipathic structure enabling 250 their interactions with class B scavenger receptors including CD36 , SR-BI, and SR-BII5 251 [34,38], as well as others, such as formyl-peptide receptors [39] and potentially class A 252 scavenger receptors. SAHPs demonstrated anti-inflammatory effects in models of athero- 253 sclerosis [37,40], cardiac ischemia–reperfusion injury, endothelial dysfunction [30,41,42], 254 endotoxemia, sepsis [16], and collagen-induced arthritis [43]. Another SAHP, 5A, de- 255 creased inflammatory and immune responses in a house dust mite–mediated murine 256 model of asthma [44]. As shown earlier, the SAHP, L-37pA, and ELK-B, targeting SR-B 257 receptors are effective inhibitors of inflammation induced by IT installation of LPS leading 258 to ALI21. However, SAHP’s effects on the long-lasting consequences of ALI such as 259 ARDS/ DAD, pulmonary function, and development of LF are not known ... The discussion section needs to be improved. It is necessary to be more concise in the presentation of the facts, clarifying the results obtained and comparing them with previous or similar studies. However, it is interesting to answer the questions that arise from these results, backed up by published literature.

7) In summary, we demonstrated that administration of L-37pA attenuates and pre- 308 vents the development of several damage-associated pulmonary syndromes in mice. 309 These results identify L-37pA as a potential compound that could be developed as an ad- 310 ditional therapy to treat pulmonary inflammation and dysfunction such as those with var- 311 ious pulmonary infectious and inflammatory insults. I suggest to insert this part in the conclusions section.

8) 5. Conclusions 416 Utilizing original unilateral ALI/ARDS/DAD model on mice we demonstrated that 417 intravenous injection L-37pA at various time points effectively reduced acute and chronic 418 inflammation, alveolar infiltration, alveolar wall and vascular infiltration and dysfunc- 419 tion, PF, as well as markedly improved pulmonary physiological functions and acceler- 420 ated pulmonary functional recovery by 2–3 weeks. The results of this study suggest that 421 SAHPs such as L-37pA, can be utilized as effective potent treatments of ARDS/DAD syn- 422 drome preventing pulmonary fibrosis. These results identify L-37pA as a potential com- 423 pound that could be developed as an additional therapy to treat pulmonary inflammation 424 and dysfunction such as those with various pulmonary infectious and inflammatory in- 425 sults. Please, underline the novelty of the study and the possible clinical implications.

Comments on the Quality of English Language

Minor changes of English language are required

Reviewer 2 Report

Comments and Suggestions for Authors

The authors evaluated SAHP, an L-37pA peptide, for the treatment of acute respiratory distress syndrome (ARDS) and the development of diffuse alveolar damage (DAD) or chronic pulmonary fibrosis (PF) in murine ICR models. Innovatively, the manuscript emphasizes that L-37pA reduces lung density faster, i.e. reduced pulmonary edema; improvement of physiological respiratory lung functions. The manuscript is aritten well and in standart English.

Minor remarks:

1. Add graphical abstract;

2. Introduction- to add to the peptide efficacy in other diseases

3.Mouse ARDS/DAD model (ARDS/DAD mice) and Ethics statement -to unite; rows 342-346 are redundant

4. row 348 to be change to :compounds targeted delivery and lung damages

5.row 358 to be change

6.to indicate all Chemokine and cytokine

7. Mononuclear infiltration and inflammation were assessed semi-quantitatively uti- 397 lizing a rate scale: 0 – normal; 1 – minimal expression; 2 – mild; 3 – medium; 4 – strongly; 398 and 5 – extraordinarily strong -the specified clarification is redundant, to be added below the histological figure or table

8. Statistics and data availability to be combined

9.Experimental groups and design + table have to be in mathrials and methods

10.Table 1. Experimental groups.to describe in detail, combat studies; number of animals ....

11.Figure 1. A and B pictures to be unified, presented on one line; arrows indicating exactly the changes are missing

12. to be .....Blood plasma Chemokine and cytokine levels

13.Kernogan index - explanation missed in Figure 5

14.Table 2 is boring and present in the max font

15.The figures from the supplement are very expressive and it will be good to present them directly.

16. Discussion part- to propose an exemplary mechanism of action of the protein at PF

17 The conclusion is poor 

18. Only 7 references are from the last 5 years. 

Comments on the Quality of English Language

-

Round 2

Reviewer 2 Report

Comments and Suggestions for Authors

-